# Survival Outcomes after Elective or Emergency Surgery for Synchronous Stage IV Colorectal Cancer

**DOI:** 10.3390/biomedicines10123114

**Published:** 2022-12-02

**Authors:** Ji-Yeon Mun, Ji-Eun Kim, Nina Yoo, Hyeon-Min Cho, Hyunho Kim, Ho-Jung An, Bong-Hyeon Kye

**Affiliations:** 1Department of Surgery, St. Vincent’s Hospital, The Catholic University of Korea, Suwon-si 16247, Gyeonggi-do, Republic of Korea; 2Department of Internal Medicine, St. Vincent’s Hospital, The Catholic University of Korea, Suwon-si 16247, Gyeonggi-do, Republic of Korea

**Keywords:** stage IV colorectal cancer, emergency, primary tumor resection, metastasectomy

## Abstract

Patients with stage IV colorectal cancer (CRC) who have not undergone primary tumor resection (PTR) are at risk of sudden medical emergencies. Despite the ongoing controversy over the necessity and timing of PTR in patients with stage IV CRC, studies comparing the survival outcomes of elective and emergency surgery in this population are scarce. This is a retrospective study conducted at a single institute. The patients were divided into two groups: the elective surgery (ELS) group (n = 46) and the emergency surgery (EMS) group (n = 26). The primary outcome was 2-year overall survival (OS). During a median follow-up period of 27.0 months, the 2-year OS was significantly better in the ELS group (80% vs. 42.9%, *p* = 0.002). No significant differences were observed in the 2-year relapse-free survival and 30-day postoperative complication rates. Planning and performing elective surgery could help increase the survival rate of patients with synchronous stage IV CRC, especially those that undergo simultaneous or staged metastasectomy.

## 1. Introduction

Approximately 25% of patients with colorectal cancer (CRC) are initially diagnosed with stage IV disease [1]. Patients with stage IV CRC present with heterogeneous clinical features [2], meaning that the treatment strategies required are diverse, complex, and influenced by several factors. Regarding surgical treatment, only a limited number of patients undergo surgery with curative intent. The majority have unresectable stage IV disease [3,4]. While surgery may be necessary for symptomatic stage IV CRC [5], its role and optimal timing in patients with unresectable stage IV CRC are controversial [6,7]. Even in patients with resectable metastasis, the timing of surgery for the primary and metastatic lesions has been debated.

Patients with unresected stage IV CRC frequently experience disease-related emergencies, such as bowel obstruction, perforation, and bleeding [8,9,10]. In addition, chemotherapy and targeted treatments may cause emergency events, such as bowel perforation and bleeding [11]. According to the SEER database, despite advances in surgical techniques and chemotherapy, the 5-year relative survival rate is only 15.1% [12]. Within this limited lifespan, preemptive PTR to prevent bowel complications should be carefully performed after weighing the pros and cons. Therefore, several studies have evaluated the survival benefit of preemptive PTR. Some advocate preemptive PTR to prevent tumor-related symptoms and increase the effectiveness of chemotherapy by reducing the tumor burden [13,14]. However, other studies have found that primary tumor-related intestinal complications during chemotherapy occur in only 13–20% of patients and have failed to show a survival benefit with preventive surgery [15,16]. Comparing the survival outcomes of elective and emergency PTR would be an important reference in determining the usefulness of preemptive PTR. However, there are few studies comparing the survival outcomes after elective and emergency surgery for stage IV CRC. We hypothesized that there would be a difference in perioperative and survival outcomes between patients undergoing planned or unplanned surgery. Hence, we aimed to compare the immediate postoperative results and survival outcomes of elective and emergent PTR in patients with stage IV CRC and determine whether elective PTR, with or without metastasectomy, is associated with a better prognosis.

## 2. Materials and Methods

(1)IRB

After obtaining approval from the Institutional Review Board of St. Vincent Hospital at the Catholic University of Korea (VC22RISI0200), we retrospectively reviewed the patients’ data and clinical information.

(2)Study design and patients

Between January 2017 and December 2019, 86 patients were initially diagnosed with synchronous metastatic CRC at St. Vincent Hospital, Catholic University of Korea. We excluded 14 patients who did not undergo PTR for CRC. Finally, 72 patients were eligible for this study and were divided into two groups: elective surgery (ELS) and emergency surgery (EMS).

(3)Definitions

Emergency surgery was defined as an operation performed because of cancer obstruction, bowel perforation, or bleeding symptoms. Elective surgery was defined as an operation performed based on a previously set treatment strategy.

(4)Elective group

All patients who were pathologically diagnosed with colorectal adenocarcinoma underwent staging work-up with abdominopelvic and chest computed tomography (CT). Liver magnetic resonance imaging (MRI) was performed when a suspicious liver lesion was observed on abdominopelvic CT (APCT). Positron emission tomography-CT was performed in patients diagnosed with extrahepatic metastatic lesions on APCT. After cancer evaluation, the treatment plan was determined by a multidisciplinary team consisting of surgeons, oncologists, radiologists, and other related specialists. 

The patients in the elective group were classified according to whether they underwent chemotherapy or surgery first. Patients with unresectable metastatic lesions or oligometastasis initially underwent chemotherapy. In addition, even in patients with resectable metastases, chemotherapy was administered to reduce their size, determine responsiveness to chemotherapy, and avoid local therapy for patients with early disease.

Patients underwent surgery first when they had almost complete intestinal obstruction, despite the absence of symptoms, or if an R0 (locally or totally) resection could be achieved. All patients were routinely treated with postoperative chemotherapy (FOLFOX or FOLFIRI regimen), with or without targeted agents, except those who refused chemotherapy or were unable to undergo treatment due to a poor general condition or underlying disease. Postoperative follow-up evaluations, including carcinoembryonic antigen (CEA) and imaging studies, were performed every 3 cycles of chemotherapy.

(5)Emergency group

Patients in the emergency group were divided into two subgroups: those who presented to an emergency room with symptoms related to CRC before being formally diagnosed, and those who developed symptoms during chemotherapy.

Patients who visited the emergency room due to an initial presentation of CRC underwent emergency resection of the primary lesion without a histological diagnosis. The pathological diagnosis was confirmed using surgical specimens.

Staging workup was performed with APCT performed in the emergency room, chest CT, and other imaging studies after the pathological diagnosis. Patients with a sudden onset of symptoms during chemotherapy were referred to a surgeon for emergency intervention. After surgery and postoperative recuperation, chemotherapy was restarted using the same regimen or a 2nd line regimen, depending on the patient’s condition and tumor status. Follow-up evaluation was performed every 3 cycles of chemotherapy, in the same manner as in the elective group.

(6)Surgery

The choice of surgical procedure depended on the site of the primary tumor. At our institution, we perform central vascular ligation (CVL) with complete mesocolic excision (CME) for colon cancer, and total mesorectal excision (TME) or tumor-specific mesorectal excision (TSME) for rectal cancer. We performed an extended radical resection if the primary tumor lesion had invaded the adjacent organs. After resection of the primary lesion, restoration of bowel continuity was performed, except in patients with septic shock, fecal peritonitis, or preoperative organ failure. If microscopic tumor infiltration of the distal or circumferential margin was observed on postoperative pathological examination, the operation was defined as an R1 resection in which R0 resection was intended. Metastasectomy was performed either simultaneously or in stages.

(7)Outcome

The primary outcome of this study was the 2-year overall survival (OS) in the ELS and EMS groups. We calculated the overall survival period from the date of the diagnosis of synchronous stage IV CRC to the date of death or end of follow-up. Relapse-free survival (RFS) was defined as the period from the start of treatment (surgery or chemotherapy) to the detection of disease progression or recurrence. The immediate postoperative outcomes (30-day complication and mortality rates) were analyzed as secondary outcomes.

(8)Statistical analyses

Statistical analysis was performed using the SPSS software, version 28.0 (IBM SPSS Statistics^®^, Armonk, NY, USA). Fisher’s exact test was used to compare categorical data and Student’s *t*-test was used for continuous data. The results are presented as mean values and standard deviations (SD) for continuous normally distributed variables, and as counts and percentages for categorical data. Survival analysis was performed using the Kaplan–Meier (log-rank test) method. Univariate and multivariate analyses of factors associated with 2-year OS were performed using the following variables: age, sex, ASA score, elective or emergency surgery, metastasectomy, and postoperative chemotherapy. Statistical significance was set at *p* < 0.05.

## 3. Results

A total of 72 patients who underwent PTR were included in this study (flow diagram shown in Figure 1). Among them, 26 patients underwent emergent CRC resection because of obstruction (23 patients) and perforation (3 patients). We compared the patients’ demographic and pathological data between the elective and emergency groups. The ASA scores (*p* = 0.015) and initial CEA levels (*p* = 0.006) were significantly higher in the EMS group. The proportion of patients who underwent preoperative chemotherapy did not differ between the ELS and EMS groups; however, the number of chemotherapy cycles administered before surgery was greater in the EMS group (*p* = 0.006) (Table 1).

There were significant differences in terms of primary tumor pathological (*p*) T classification, where pT4 tumors (*p* = 0.015) and multiorgan metastases (*p* = 0.005) were significantly more common in the EMS group than in the ELS group. Resectable metastatic lesions were more common (*p* = 0.003) and metastasectomy was performed more frequently in the ELS group (*p* < 0.001). Although there was no difference in the radicality of the primary tumor lesion, R0 status, including for the metastatic lesions, was significantly more common in the ELS group (Table 2). There was no difference in 30-day postoperative complication rates between the two groups (Table 3). At a median follow-up of 27.0 months, the 2-year OS rate was higher in the ELS group (80% vs. 42.9%, *p* = 0.002), but there was no difference in the 2-year RFS rate (Figure 2).

Since the two groups differed with respect to initial resectability, we performed a subgroup analysis of 2-year OS according to initial resectability. Among patients with initially resectable metastatic lesions, the 2-year OS rate was significantly better in the ELS group (86.21% vs. 34.29%, *p* = 0.01). Among patients with unresectable lesions, the ELS group showed better 2-year OS, but the difference was not statistically significant (75.49% vs. 46.44%, *p* = 0.098) (Figure 3).

Table 4 shows the results of the univariate and multivariate Cox hazard analyses for 2-year OS. Emergent surgery (*p* = 0.002), higher ASA score (*p* < 0.001), no metastasectomy (*p* = 0.006), and postoperative chemotherapy (*p* = 0.014) were associated with worse 2-year OS. In the multivariate analysis, only a higher ASA score (*p* = 0.002) and no metastasectomy (*p* = 0.018) remained significant risk factors. In the univariate analysis of 2-year RFS, only the administration of postoperative chemotherapy (*p* = 0.043) was a statistically significant factor (Table A1).

We performed an additional subgroup analysis for survival outcomes between patients who underwent elective surgery as the first treatment (n = 32) and those who underwent emergency surgery while receiving chemotherapy (n = 8). There was a trend towards better 2-year OS in the elective group (74.3% and 37.5%), but the difference was not statistically significant (*p* = 0.052) (Figure 4). We also compared the effect of postoperative chemotherapy on patient survival. Of the 72 patients, 50 had received postoperative chemotherapy, with no significant difference in the rate of postoperative chemotherapy between the ELS and EMS groups (76.1% vs. 57.7%, *p* = 0.104). The patients were accordingly divided into four subgroups according to the treatment received: ELS with/without postoperative chemotherapy and EMS with/without postoperative chemotherapy. The subgroup of patients who underwent elective surgery followed by chemotherapy exhibited the best 2-year OS (*p* < 0.001) (Figure 5).

We also analyzed stoma-free survival between the ELS and EMS groups. Stoma formation was performed in 19 patients in the ELS group and 4 in the EMS group. Among these, stoma-takedown surgery was performed in 63.2% of patients in the ELS group and 50% of those in the EMS group (Table 5). The ELS group showed better stoma-free survival (26.16 vs. 13.25 months), but the difference was not statistically significant (*p* = 0.701) (Figure 6).

## 4. Discussion

In patients with stage IV CRC, the primary tumor is usually at an advanced stage of T3 or higher, putting them at risk of emergencies, such as obstruction, perforation, and bleeding. Approximately 7–29% of CRC patients present with bowel obstruction [17], and the morbidity, mortality, and stoma formation rates are higher for patients requiring emergent intervention than for those managed electively [8]. However, few studies have investigated the outcomes of emergent PTR in stage IV CRC patients. Evaluating surgical outcome of stage IV CRC patients who underwent emergency PTR, this study demonstrates that a more advanced pathological T stage (T4) was more common in the EMS group, and 23 of the 26 patients underwent emergency PTR for bowel obstruction.

Surgical treatment of obstructive CRC is different depending on the location of the cancer [18]. Emergency PTR with primary anastomosis is mainly performed for right (Rt.) colon cancer obstruction. However, with the development of colonoscopy, staged operation after self-expandable metal stent (SEMS) can be considered in Rt. colon cancer obstruction [19], if the patient is stable and the cancer is not located at the IC valve or cecum. More diverse treatment can be considered in left (Lt.) colon cancer obstruction than in Rt. Colon cancer. Hartmann’s procedure is the most preferred surgical treatment in high-risk patients with Lt. colon cancer obstruction [20]. Fecal diversion without PTR could be another option. If the patient’s vital sign is stable and there is no sign of impending perforation or ischemic change, staged operation after SEMS is recommended [21,22,23].

In this study, patients with Rt. Colon and proximal Lt. sided colon cancer obstruction underwent PTR without a trial of SEMS. In patients with distal Lt. sided colon cancer obstruction, PTR and primary anastomosis was performed when the patient’s vital sign and bowel condition were acceptable. Moreover, PTR with or without anastomosis was performed inevitably due to the failure of the SEMS or ischemic change of the bowel.

Although the difference was not statistically significant, it is notable that there were two local R2 resections in the EMS group compared to none in the ELS group. Furthermore, the colostomy rate was high, and the stoma closure rate was low in the EMS group, even though the ELS group included 4 APR cases. Based on these results, it is possible to prevent local R2 resection and perform simultaneous metastasectomy by staged operation after SEMS insertion or planned surgery if the primary tumor is resectable.

Patients who underwent elective surgery showed a significantly better 2-year OS than those who underwent emergency surgery. As the initial resectability rates differed between the ELS and EMS groups, we compared the 2-year OS between the two groups according to resectability. Although statistical significance was confirmed only in patients with initially resectable tumors, the ELS group showed improved 2-year OS for both initially resectable and unresectable metastatic lesions. This result supports the possibility that elective surgery contributes to a better survival outcome, regardless of the initial resectability of the metastatic lesion. In the multivariate analysis, ASA score and whether metastasectomy was performed were the most significant factors associated with survival. Patients in the ELS group showed better ASA scores, more resectable metastatic lesions, and underwent more metastasectomies than those in the EMS group. Several retrospective and prospective studies on stage IV CRC have demonstrated that curative metastasectomy improves survival and may even be curative [24,25]. The results of this study are consistent with these previous findings.

In asymptomatic patients with stage IV CRC, the benefit of PTR as the initial treatment is controversial. While it may reduce the tumor burden, previous studies have shown that PTR increases postoperative morbidity and mortality rates without improving survival [12,26]. However, unresected primary tumor can also cause emergent events that start chemotherapy first. Therefore, we performed a subgroup analysis to compare asymptomatic patients who underwent elective surgery first with those who underwent emergency surgery while receiving chemotherapy. Although the difference was not statistically significant, asymptomatic patients who underwent elective surgery as the initial treatment tended to have improved survival rates.

Regarding postoperative chemotherapy, there was no significant difference in its frequency, its timing after surgery, and the number of cycles between the ELS and EMS groups. However, the univariate analysis showed that the administration of postoperative chemotherapy was significantly associated with OS, and was the only factor associated with RFS. Furthermore, in the subgroup analysis, patients who received chemotherapy after elective surgery showed significantly better survival outcomes than other cohorts. Hirotoshi et al. demonstrated that adjuvant chemotherapy improved OS after curative resection for stage IV CRC [27]. In the ELS group, R0 metastasectomy was more common than in the EMS group. Thus, the results of the subgroup analysis with respect to postoperative chemotherapy are consistent with previous findings.

As the clinical outcome for patients with stage IV CRC has improved [28], their quality of life (QOL) has become an increasingly important consideration. In a UK survey [29] of CRC survivors, colostomy was clearly associated with reduced QOL. Therefore, we analyzed stoma-related outcomes, finding that while there was no difference in stoma closure rate between the groups, the ELS group showed a trend towards better stoma-free survival. While the difference in survival was not significant, given the small number of patients who underwent stoma takedown surgery in the EMS group, it is still notable considering that four patients in the ELS group had undergone abdominoperineal resections.

This study has some limitations. First, this was a retrospective, non-randomized, study conducted at a single center with a small number of patients. Since there is no consensus on the superiority of simultaneous or staged resection of metastatic lesions [3,30,31], we determined the timing of metastasectomy individually for each patient after multidisciplinary consultation. Finally, the fact that the clinicopathological characteristics of patients were worse in the EMS group may have affected their poor survival outcomes.

The strength of this study is that it is one of very few that have investigated the issue of survival after emergency surgery for stage 4 CRC. We have also analyzed the factors associated with stoma-free survival and the influence of chemotherapy. Additionally, emergency colorectal surgery is often performed by unspecialized general surgeons, which can reportedly influence the outcome of surgery [32]. However, in this study, surgical quality was guaranteed because all PTRs were performed by a specialized colorectal surgeon.

## 5. Conclusions

In conclusion, in patients who are fit for surgery, elective PTR showed better survival outcomes, especially when performed with simultaneous or staged metastasectomies. However, the benefit of elective PTR alone and surgery in asymptomatic patients remains unclear. Future prospective, observational, and large-scale studies are required to confirm these encouraging results. In addition, careful patient selection is needed to identify those who stand to benefit the most from PTR.

## Figures and Tables

**Figure 1 biomedicines-10-03114-f001:**
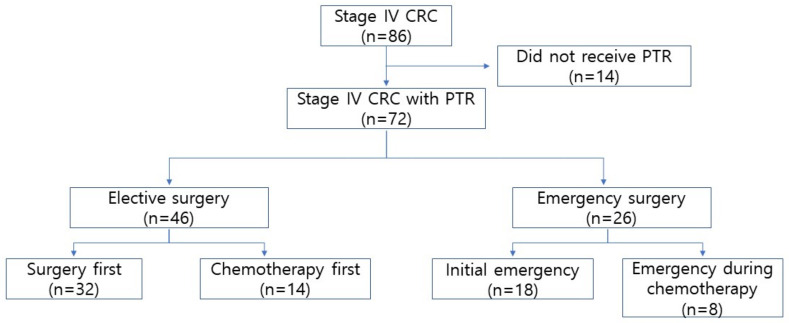
Flow diagram CRC, colorectal cancer; PTR, primary tumor resection.

**Figure 2 biomedicines-10-03114-f002:**
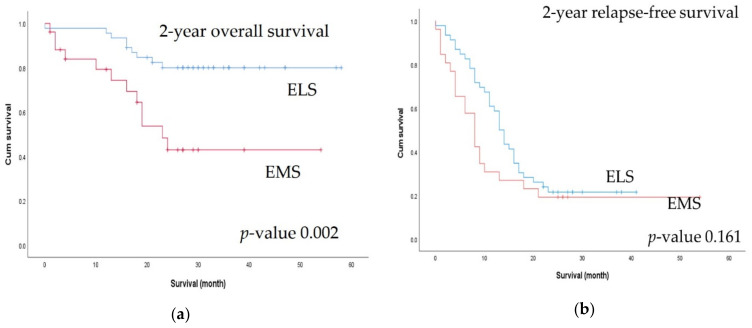
Comparison of 2-year survival outcomes between the elective surgery (ELS) and emergency surgery (EMS) groups: (**a**) 2-year overall survival; (**b**) 2-year relapse-free survival.

**Figure 3 biomedicines-10-03114-f003:**
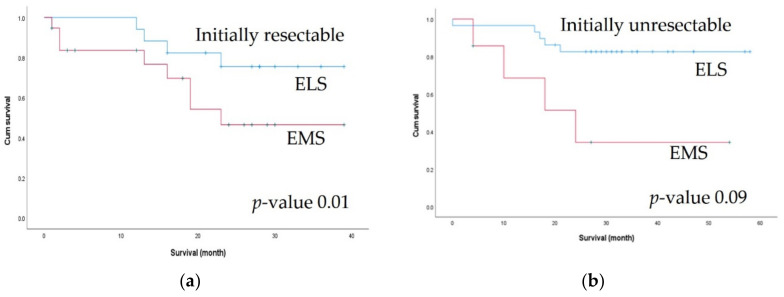
Comparison of 2-year survival outcomes between the elective surgery (ELS) and emergency surgery (EMS) groups according to initial resectability: (**a**) 2-year overall survival in initially resectable patients; (**b**) 2-year overall survival in initially unresectable patients.

**Figure 4 biomedicines-10-03114-f004:**
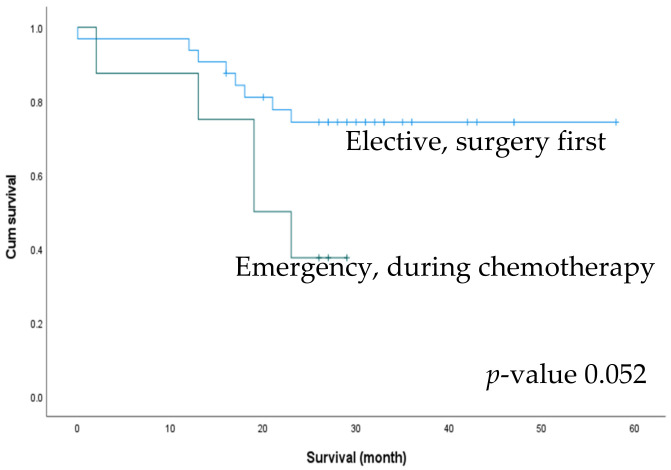
Comparison of 2-year survival outcomes between surgery first patients in the elective group and patients with an emergent event during chemotherapy.

**Figure 5 biomedicines-10-03114-f005:**
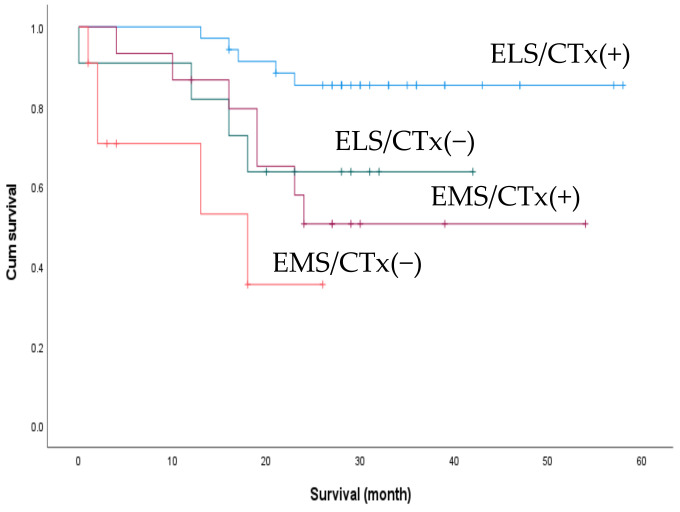
C Comparison of 2-year overall survival between patients who received postoperative adjuvant chemotherapy (post-op CTx) and those who did not. ELS/CTx(+), elective surgery + post-op CTx; ELS/CTx(−), elective surgery + no post-op CTx; EMS/CTx(+), emergency surgery + post-op CTx; EMS/CTx(−), emergency surgery + no post-op CTx.

**Figure 6 biomedicines-10-03114-f006:**
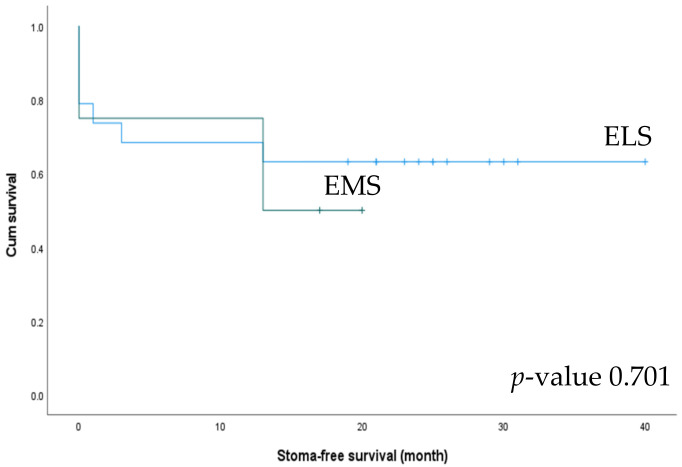
Stoma-free survival of patients undergoing left-side colectomy in the elective surgery (ELS) and emergency surgery (EMS) groups.

**Table 1 biomedicines-10-03114-t001:** Baseline characteristics.

Title 1		ELS (n = 46)	EMS (n = 26)	*p*-Value
Sex (M/F)		29 (63%)/17	16 (61.5%)/10	0.899
Age (years)		63.22 ± 12.02	66.96 ± 13.53	0.266
BMI (kg/m^2^)		23.30 ± 3.53	21.58 ± 2.79	0.238
ASA score	I/II	44 (95.7%)	20 (76.9%)	0.015
	III/IV	2 (4.3%)	6 (23.1%)	
Initial CEA (ng/mL)		62.42 ± 166.95	164.75 ± 295.87	0.006
Preoperative chemotherapy	Yes	14 (30.4%)	8 (30.8%)	0.976
	No	32 (69.6%)	18 (69.2%)	
Preoperative chemotherapy cycle		10.93 ± 1.82	16.13 ± 0.99	0.006
Operation for primary tumor	RHC	7 (15.2%)	11 (42.3%)	0.009
	LHC	2 (4.4%)	3 (11.5%)	
	AR	10 (21.7%)	4 (15.3%)	
	LAR	17 (37.0%)	1 (3.8%)	
	ISR	3 (6.5%)	0	
	APR	4 (8.7%)	0	
	Hartmann	1 (2.2%)	3 (11.5%)	
	STC/TC	2 (4.4%)	4 (15.4%)	

APR, abdominoperineal resection; AR, anterior resection; ASA, American Society of Anesthesioogists; BMI, body mass index; CEA, carcinoembryonic antigen; ELS, elective surgery; EMS, emergency surgery; ISR, intersphincteric resection; LAR, low anterior resection; LHC, left hemicolectomy; RHC, right hemicolectomy; STC, subtotal colectomy; TC, total colectomy.

**Table 2 biomedicines-10-03114-t002:** Tumor characteristics.

Title 1		ELS (n = 46)	EMS (n = 26)	*p*-Value
Location of primary tumor (sidedness)	Right	8 (17.4%)	12 (46.2%)	0.009
	Left	38 (82.6%)	14 (53.8%)	
Location of primary tumor	Colon	30 (65.2%)	24 (92.3%)	0.011
	Rectum	16 (34.8%)	2 (7.7%)	
Pathology T stage	pT2/T3	20 (43.5%)	4 (15.4%)	0.015
	pT4	26 (56.5%)	22 (84.6%)	
Pathology N stage	pN0	8 (17.4%)	2 (7.7%)	0.200
	pN1	21 (45.7%)	9 (34.6%)	
	pN2	17 (37.0%)	15 (57.7%)	
Radicality of the primary lesion	R0	40 (87.0%)	21 (80.8%)	0.162
	R1	6 (13.0%)	3 (11.5%)	
	R2	0	2 (7.7%)	
Radicality of all lesions	R0	25 (54.3%)	7 (26.9%)	0.028
	R1	2 (2.8%)	0	
	R2	19 (41.3%)	19 (73.1%)	
Metastatic lesion	Single organ	40 (87.0%)	15 (57.7%)	0.005
	Multiple organs	6 (13.0%)	11 (42.3%)	
Initial resectability of metastatic lesion	Resectable	29 (63.0%)	7 (26.9%)	0.003
	Non-resectable	17 (37.0%)	19 (73.1%)	
Metastasectomy	Yes	34 (73.9%)	8 (30.7%)	<0.001
	No	12 (26.1%)	18 (69.2%)	

**Table 3 biomedicines-10-03114-t003:** Perioperative outcomes.

		ELS (n = 46)	EMS (n = 26)	*p*-Value
Operative time (min)		293.33 ± 108.18	220.19 ± 55.01	0.001
Operative approach	Laparoscopic	25 (54.3%)	5 (19.2%)	0.004
	Open	21 (45.7%)	21 (80.8%)	
Postoperative hospital stay (days)		11.85 ± 8.33	12.54 ± 9.41	0.411
Postoperative complication		24 (52.2%)	8 (30.8%)	0.079
Severe complication		5 (10.9%)	2 (7.7%)	0.662
Reoperation (%)		2 (4.3%)	0	0.281
Mortality within 30 days (%)		1 (2.2%)	0	0.449
Postoperative chemotherapy	Yes	35 (76.1%)	15 (57.7%)	0.104
	No	11 (23.9%)	11 (42.3%)	
Postoperative chemotherapy cycle		10.74 ± 1.43	14.60 ± 2.62	0.398
Interval between PTR and start of chemotherapy (days)		44.23 ± 6.49	48.53 ± 9.33	0.649

ELS, elective surgery; EMS, emergency surgery; PTR, primary tumor resection.

**Table 4 biomedicines-10-03114-t004:** Analysis of factors associated with 2-year overall survival.

Factor	N	2-Year OS (%)	Univariate Analysis	Multivariate Analysis	
			*p*-Value	HR (95% CI)	*p*-Value
Surgery					
Elective	46	82.39	0.002		0.177
Emergency	26	42.94			
Age (years)					
<65	35	71.11	0.512		0.641
>65	37	65.79			
Sex					
Male	45	66.34	0.759		0.579
Female	27	71.61			
BMI (kg/m^2^)					
<23	36	69.87	0.699		
>23	36	66.69			
ASA score					
I-II	64	73.17	<0.001	Ref.	0.002
III-IV	8	20.83		4.964 (1.771–13.91)	
Tumor sidedness					
Right	20	56.55	0.142		
Left	52	72.87			
Tumor location					
Colon	54	62.52	0.129		
Rectum	18	83.33			
Pathological T stage					
T2-3	30	72.48	0.448		
T4	42	65.28			
Pathological N stage					
N0	6	100	0.120		
N+	66	65.27			
Number of metastases					
1	55	70.59	0.478		
≥2	17	60.81			
Initial resectability					
Yes	36	74.20	0.277		
No	36	61.67			
Metastasectomy					
Yes	42	80.23	0.006	Ref.	0.018
No	30	48.76		2.920 (1.198–7.119)	
Postoperative chemotherapy					
Yes	50	75.08	0.014		0.085
No	22	52.29			

ASA, American Society of Anesthesiologists; BMI, body mass index; CI, confidence interval; HR, hazard ratio; OS, overall survival.

**Table 5 biomedicines-10-03114-t005:** Stoma creation in patients with left-sided cancer.

		ELS (n = 38)	EMS (n = 14)	*p*-Value
Creation of stoma	No	19 (50.0%)	10 (71.4%)	0.076
	Ileostomy	11 (28.9%)	0	
	Colostomy	8 (21.1%)	4 (28.6%)	
Operation performed in patients with a stoma	LAR	10 (52.6%)	2 (50%)	0.043
	ISR	3 (15.7%)	0	
	Hartmann’s	1 (5.3%)	2 (50%)	
	STC	1 (5.3%)	0	
	APR	4 (21.1%)	0	
Stoma closure		12/19 (63.2%)	2/4 (50%)	0.624

APR, abdominoperineal resection; ELS, elective surgery; EMS, emergency surgery; ISR, intersphincteric resection; LAR, low anterior resection; SF, splenic flexure; STC, subtotal colectomy. Left-sided cancer was defined as tumors located between the splenic flexure and the colon.

## Data Availability

The datasets during and/or analyzed during the current study are available from the corresponding author on reasonable request.

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
