# Peer review of "Survival Outcomes after Elective or Emergency Surgery for Synchronous Stage IV Colorectal Cancer"

_biomedicines, 2022, doi:10.3390/biomedicines10123114_

Round 1
Reviewer 1 Report
Comments
The manuscript by Ji Yeon Mun et al. is an interesting paper dealing with the evaluation of survival outcomes of patients with stage IV colorectal cancer after elective or emergency surgery. In detail, authors aimed to contribute addressing the controversy arising from the necessity and timing of the primary tumor resection. The topic of the paper is worthy of investigation, well fits with the scope of the journal, and is of great interest for scientists working in the field.
It is an opinion of this reviewer that the manuscript should be eventually accepted for publication. Some improvements are suggested below:
Introduction: authors should improve the discussion about the controversy within the literature data cited at lines 44-45.
Materials and methods: no specific comments to this section
Results: no specific comments to this section
Discussion: authors should discuss their results among with the available literature data to strengthen the impact of their study.
Minor points: Please check format of Figures and Tables (e.g. legend of Figure 1 and footnotes of Table 1)
Author Response
- Introduction: authors should improve the discussion about the controversy within the literature data cited at lines 44-45.
“Although there are ongoing studies on the necessity and timing of the primary tumor resection (PTR), the results are still controversial.” -
We deleted this phrase because the previous paragraph already introduced the controversy of the PTR. Instead, we added reference for why the controversy over PTR in stage IV is important at lines 36-40.
- Discussion: authors should discuss their results among with the available literature data to strengthen the impact of their study.
We included the topic of colonic obstruction in the discussion because it is the main reason for emergency surgery in CRC. We added a paragraph about surgical strategies according to a location in obstructive CRC (253-262). Also, we described surgical treatments performed for emergency obstructive CRC in this study (263-273).
- Minor points: Please check format of Figures and Tables (e.g. legend of Figure 1 and footnotes of Table 1)
We changed the format of the legends of all figures and footnotes of all tables. Also, we changed figure 1 to a more high-resolution image.
Reviewer 2 Report
The paper is an original research comparing the outcomes in terms of 2-years overall survival. The article is well structured and based on solid statistical analysis. However, the study groups are small, but this was acknowledged bt the authors as a limitation of the study.
Some minor issues:
1. In the discussion section, please add a paragraph regarding the surgical strategy for occlusive colorectal cancer in emergency: one step vs 2steps surgery
2. The conclusions should be better focused on the comparative results between the patients operated in emergency vs elective surgery.
Author Response
- In the discussion section, please add a paragraph regarding the surgical strategy for occlusive colorectal cancer in emergency: one step vs 2steps surgery
It was an excellent decision to include this topic in the discussion because obstruction is the main reason for emergency surgery in CRC. Therefore, we added a paragraph about surgical strategies according to a location in obstructive CRC (253-262). Also, we described surgical treatments performed for emergency obstructive CRC in this study (263-273).
- The conclusions should be better focused on the comparative results between the patients operated in emergency vs elective surgery.
We agree with the reviewer and correct the phrase of the conclusion to focus on the comparative results (329-330).
Round 2
Reviewer 1 Report
Authors well addressed all the concerns and the manuscript was improved. This reviewer is recommending publication of the current version of the manuscript.